# Distinction Between Proliferative Lupus Nephritis and Membranous Lupus Nephritis Based on Inflammation, NETosis, and Glomerular Exostosin

**DOI:** 10.3390/ijms26188769

**Published:** 2025-09-09

**Authors:** Yukihiro Wada, Hiroyuki Okawa, Tetsuya Abe, Kazuhiro Takeuchi, Mariko Kamata, Emiko Takeuchi, Tadahiro Suenaga, Masayuki Iyoda, Yasuo Takeuchi

**Affiliations:** 1Department of Nephrology, Kitasato University School of Medicine, Sagamihara 252-0374, Kanagawa, Japan; hiro@insti.kitasato-u.ac.jp (H.O.); tetsuyaa@med.kitasato-u.ac.jp (T.A.); takekazu04044@yahoo.co.jp (K.T.); ytakeuch@med.kitasato-u.ac.jp (Y.T.); 2Department of Pharmacology, Kitasato University School of Medicine, Sagamihara 252-0374, Kanagawa, Japan; m.kamata@med.kitasato-u.ac.jp; 3Department of Immunology, Kitasato University School of Medicine, Sagamihara 252-0374, Kanagawa, Japan; emichike@med.kitasato-u.ac.jp (E.T.); suenaga.tadahiro@kitasato-u.ac.jp (T.S.); 4Department of Microbiology and Immunology, Showa Medical University School of Medicine, Tokyo 142-8555, Japan; iyoda@med.showa-u.ac.jp; 5Division of Nephrology, Department of Medicine, Showa Medical University School of Medicine, Tokyo 142-8555, Japan

**Keywords:** proliferative lupus nephritis, membranous lupus nephritis, NETosis, EXT1/EXT2

## Abstract

Lupus nephritis (LN) is a serious complication of systemic lupus erythematosus that is associated with long-term morbidity and mortality. Pathomorphological findings of LN are broadly divided into proliferative lupus nephritis (PLN) and membranous lupus nephritis (MLN). PLN is characterized by diffuse global or segmental proliferative glomerulonephritis with significant infiltration of inflammatory cells. Type 1 T-helper (Th1) cells, which predominate under inflammatory conditions, and NETosis, as the process of forming neutrophil extracellular traps (NETs), are key factors in the development of PLN. Meanwhile, MLN is characterized by diffuse membranous nephropathy (MN) with global granular subepithelial immune deposits. MLN patients usually experience massive proteinuria, and occasionally show an unfavorable renal prognosis despite aggressive treatment, similar to PLN patients. Intriguingly, in some instances, MLN patients do not show the general immunoserological characteristics of SLE, such as low serum complement and elevated anti-DNA antibody titers. Several reports have indicated an association between Th2 cell dominance and the development of MLN. Moreover, exostosin 1 (EXT1) and exostosin 2 (EXT2) on the glomerular basement membrane have recently been discovered as novel putative antigens for secondary MN, and have been shown to be up-regulated in patients with MLN. To date, many studies have focused on the dissimilarities between PLN and MLN. However, the reason for two polar morphological forms existing within the same disease is not completely clear. The present review addresses published observations on this topic in addition to providing our assertion regarding characteristic NETosis and glomerular EXT1/EXT2 expressions between PLN and MLN.

## 1. Introduction

Systemic lupus erythematosus (SLE) is a chronic multisystem autoimmune disease characterized by a disturbed interplay between the innate and adaptive immune systems with a loss of self-tolerance and overproduction of autoantibodies [1,2,3,4]. Nowadays, the estimated worldwide prevalence and annual incidence of SLE are 2.9–241 patients per 100,000 people and 0.3–23.2 patients per 100,000 person-years, respectively [5,6]. SLE mainly affects female patients, while the prevalence and incidence are considered to depend on multiple factors such as age, sex, genetics, ethnicity, environmental factors, and immunoregulatory abnormalities [7,8]. In clinical settings, patients with SLE present with variable disease severity and tend to have multiple organ involvement. For example, SLE patients may exhibit only fatigue and mild skin lesions or life-threatening manifestations such as neuropsychiatric, hematological, cardiovascular or renal disorders [1,6]. The hallmark and essential points of SLE are activation or amplification of inflammatory cells, including neutrophils, macrophages, dendritic cells, and lymphocytes, in addition to the production of a variety of autoantibodies against various antigens derived from the cell membrane (such as antiphospholipid and anticardiolipin autoantibodies), cytosol (such as anti-SSA and anti-SSB autoantibodies) or nucleus (anti-double-stranded DNA autoantibodies) [1,6]. Moreover, intriguingly, patients with SLE have been demonstrated to carry autoantibodies against mitochondrial components and infectious pathogens such as virus components [9,10,11,12]. As a result of autoantibodies binding to those different planted antigens, immune complexes (ICs) form and are deposited in different tissues and organs, leading to various clinical manifestations.

Renal involvement, termed lupus nephritis (LN), is a serious complication of SLE associated with long-term morbidity and mortality [1,13]. Representative histological findings of LN include proliferative lupus nephritis (PLN) and membranous lupus nephritis (MLN), which are recognized as two polar morphological forms [14,15]. According to previous reports, PLN is characterized by diffuse global or segmental proliferative glomerulonephritis, and patients with PLN tend to show unfavorable renal prognosis despite aggressive treatment [8,15]. In contrast, MLN is characterized by diffuse membranous nephropathy with global granular subepithelial immune deposits [15], and MLN patients usually show massive proteinuria and an indolent clinical course [14,16]. Moreover, intriguingly, ordinal immunoserological characteristics of SLE, such as low serum complement and elevated anti-DNA antibody titers, are not obvious in a proportion of MLN patients [14,17]. In accordance with the recently reported classification for LN by the International Society of Nephrology/Renal Pathology Society (ISN/RPS) [15], glomerulonephritis containing class III and IV lesions can be recognized as PLN; meanwhile, pure class V lesions can be categorized as MLN. Thus far, several reports have indicated that such histological variance could be due to imbalances in T-helper (Th) cell-related cytokines [16,17,18,19,20,21]. However, the reason for two polar morphological forms being present in the same disease is not completely clear. In clinical practice, PLN and MLN frequently coexist as a combined disease. Moreover, these two types of LN occasionally appear independently in the same patient as distinct presentations [22]. Thus, elucidating the essential factors that determine the morphological form of LN is important.

Recently, deep involvement of neutrophil extracellular traps (NETs) in the pathogenesis of inflammatory renal diseases such as LN and anti-neutrophil cytoplasmic antibody (ANCA)-related glomerulonephritis has been highlighted [23,24,25,26]. Persistently activated NETs without proper procession have been proposed to induce inflamed PLN [25]. Meanwhile, the proteins exostosin 1 and exostosin 2 (EXT1/EXT2) on the glomerular basement membrane (GBM) have been reported as novel causative antigens in patients with MLN [22,27,28]. These advances in knowledge regarding the pathogenesis of PLN or MLN might offer an intriguing clue to distinguishing the morphological forms of LN or uncovering the characteristic developmental mechanisms for each lesion in LN. However, to the best of our knowledge, such analyses have not yet been sufficient. This review highlights recent insights into PLN and MLN by comparing NETosis profiles and glomerular EXT1/EXT2 expression, including our previously published analysis, and proposes potential distinctions in the pathogeneses.

## 2. Method for Literature Search

A comprehensive literature search was conducted using PubMed (MEDLINE) to identify relevant publications related to LN. The search included articles published between 1 January 2010 and 31 July 2025. The full search strategy for PubMed was as follows: (“Lupus” [MeSH Terms] OR “lupus” [Title/Abstract]) AND (“inflammation” [Title/Abstract] OR “lymphocyte [Title/Abstract]). Additional searches were conducted using combinations such as “lupus” AND “membranous nephropathy”, “lupus” AND “neutrophil extracellular traps”, and “lupus” AND “exostosin”. All retrieved references were imported into the EndNote program (Version 20), and duplicate records were removed automatically using the software’s deduplication function.

Regarding study selection process, the initial screening of titles and abstracts was primarily performed by the first author. Subsequent evaluation of the full texts, selection of relevant studies, and assessment of their scientific value were carried out in collaboration with all co-authors to ensure accuracy, objectivity, and consistency. Any disagreements regarding the inclusion of articles were resolved through discussion and consensus among the authors. Although the main focus was on literature published after 1 January 2010, earlier studies were also considered for inclusion if they were deemed highly reliable and relevant to this review article. Such references were included only with the agreement of all co-authors. In terms of inclusion and exclusion criteria, original peer-reviewed articles written in English, involving human studies, and addressing the pathophysiology, biomarkers, pathological findings or treatment strategies related to LN. Additionally, case reports and review articles (including those without original data) were considered for inclusion if they were judged to be of particular value to this review article, and were included with the consensus of the authors. Publications that were conference abstracts or non-English language articles were excluded from this review article.

## 3. Pathomorphological Differences Between PLN and MLN

In 2004, the pathological diagnosis of LN was categorized into Classes I–VI based on the ISN/RPS criteria, which focus on glomerular lesions [29,30]. Thereafter, Bajema et al. revised the interpretation of the classification for glomerular lesions in LM (2018 ISN/RPS classification) [15]. In the developmental mechanisms of LN, glomerular deposition of immunoglobulin (Ig)G-related immune complexes (ICs) is indispensable, representing a feature that is not typical in other autoimmune diseases such as rheumatoid arthritis, systemic sclerosis, polymyositis/dermatomyositis, and microscopic polyangiitis. In patients with SLE, anti-double-stranded (ds) DNA antibody (Ab) and anti-Smith antigen (Sm) Ab form antigen-Ab ICs consisting of IgG and complement, which are then deposited in the mesangial area and subendothelial space of the GBM [31]. In cases with only IgG1 deposition in glomeruli, complement activation and glomerular damage are not particularly serious (Class I LN). However, in cases where IgG3, which shows strong complement-binding properties, is deposited in glomeruli, the classical complement pathway is activated and results in deposition of C1q, C4, and C3. This process is accompanied by a reduction in serum complement levels including C3, C4, and CH50. According to the 2018 update of the ISN/RPS classification, the glomerular lesions of class II LN are mild. Sustainable glomerular deposition of ICs and complement activation recruit inflammatory cells including neutrophils and mononuclear cells into inflamed glomeruli, resulting in amplification of mesangial cell damage, endothelial cell enlargement, endocapillary proliferative glomerular injury, GBM rupture, and podocyte injury. Such inflammatory damage to the glomeruli causes massive proteinuria and hematuria. When less than half of the glomeruli in the renal biopsy specimen are damaged and filled with active pathological findings such as wire loop lesions, endocapillary proliferative changes, and crescent lesions consisting of extracapillary hypercellularity, the LN lesion is categorized as class III. When more than half of the glomeruli in the renal biopsy specimen are damaged and filled with active pathological findings, the LN lesion is categorized as Class IV. Class V LN, equivalent to membranous nephropathy, is diagnosed with prominent spike formation and thickening of the GBM [15,31]. As mentioned above, GN with active lesions consisting of wire loop lesions, endocapillary lesions, and crescentic formation (categorized as class III or IV lesions under the 2018 ISN/RPS classification [15]) can be recognized as PLN. The pure class V lesion in the 2018 ISN/RPS classification [15] can be categorized as MLN.

In the typical renal biopsy findings in PLN case with massive proteinuria and significant hematuria, infiltration of inflammatory cells such as neutrophils and mononuclear cells within the glomerular capillaries and “wire loop” thickened capillary wall are seen under light microscopy (LM) with hematoxylin and eosin (HE) staining. This finding corresponds to endocapillary proliferative glomerulonephritis, in which both deposits and endothelial hypercellularity are confirmed under LM with periodic acid–Schiff staining or periodic acid methenamine silver (PAM) staining. Electron microscopy (EM) shows differently sized (large or small) electron-dense deposits (EDDs) in the subendothelial and paramesangial areas in glomeruli. Immunofluorescence (IF) staining shows not only significant IgG granular deposits, but also deposits for IgA and IgM, C3, C4, and C1q in glomeruli. These findings are recognized as the typical “full-house pattern” of PLN. In particular, intense glomerular positivity for IgG3, C3, and C1q is important for the diagnosis of PLN. Moreover, in patients with PLN, virus-like particles (microtubular structures) are occasionally observed on EM.

Another important pathological finding in patients with PLN is an antiphospholipid syndrome nephritis (APSN) [31], which is sometimes found in patients with SLE. Antiphospholipid syndrome (APS) is typically diagnosed in patients with positive tests for anticardiolipin antibodies and lupus anticoagulant, often accompanied by an elevated activated partial thromboplastin time [31]. APS coexists with SLE in approximately 30–40% of cases [32,33,34]. In particular, the coexistence of PLN with APSN is critical. According to the 2023 ACR/EULAR classification criteria for APS [35], APSN includes a range of vascular lesions such as thrombotic microangiopathy, fibrous intimal hyperplasia, and focal cortical atrophy, all of which are distinct from the IC-mediated glomerular lesions typically detected in PLN. Importantly, these two pathological processes can co-occur within the same kidney biopsy, resulting in overlapping and complex renal lesions. This overlap carries significant clinical implications. Patients with both PLN and APSN have been reported to exhibit poorer renal outcomes, an increased risk of thrombotic events, and reduced responsiveness to immunosuppressive therapy alone [36]. Therefore, in the management of PLN, identifying vascular lesions associated with APSN is crucial for guiding adjunctive treatment strategies, particularly anticoagulation therapy, in addition to conventional immunosuppressive therapy for PLN.

In terms of renal biopsy findings of MLN cases with nephrotic range proteinuria, typical LM findings are thickening of the GBM and prominent spike formation in the GBM. In some instances, endocapillary hypercellularity results from the influx of leukocytes and monocytes/macrophages into capillary lumens is observed. Regarding EM findings, subepithelial EDDs are common. Small amounts of EDDs are also seen in the subendothelial and paramesangial areas, which could provide a clue to differentiating between conditions with primary membranous nephropathy. IF findings showed global glomerular granular deposits of IgG, C3, and C1q. Diverse glomerular staining patterns are observed with IgG subclasses according to the causative antigen on the GBM. Basically, MLN also presents with a “full-house pattern” of IF findings caused by epithelial deposition of several immunoglobulins and complement components including C1q, which can allow differentiation from primary membranous nephropathy. However, in cases with the aforementioned atypical immunoserological characteristics of SLE, the definition of MLN in SLE is occasionally a difficult task. Causative or pathogenic antigens for MLN on the epithelia of glomeruli provide a clue for the developmental mechanisms of MLN, such as M-type phospholipase A2 receptor (PLA2R) for primary membranous nephropathy [37].

## 4. Dissimilarities Between PLN and MLN Based on T Lymphocyte-Related Abnormalities

First, properly understanding the involvement of innate and adaptive immune mechanisms in the developmental mechanisms for LN is critical. The pathogenesis of LN involves a complex interplay between innate and adaptive immune responses [3]. Innate immunity plays a pivotal role in the initiation of autoimmunity. In patients with LN, inappropriate processes of cell death and the release of NETs are considered as triggers for activation of the innate immune system. Numerous immunoregulatory factors including neutrophils, macrophages, complement pathways, type I interferons (IFNs), interleukins (ILs), and altered activity of pattern recognition receptors (PRRs), particularly Toll-like receptors (TLRs), are involved in the amplification loop for activation of the innate immune system [38,39,40]. Briefly, impaired clearance of apoptotic cells or necrotic cell debris leads to the release of endogenous nucleic acid-containing antigens. These autoantigens are recognized by PRRs on plasmacytoid dendritic cells, triggering overproduction of type I IFNs, particularly IFN-α, in turn promoting inflammation and activating lymphocytes. Neutrophils and macrophages contribute to the impaired clearance of apoptotic and necrotic cell debris, as well as the formation of NETs, which expose pathogenic nuclear antigens. Complement activation can exacerbate tissue injury when the system is dysregulated [39,40]. The processes of adaptive immunity sustain the autoimmune response. The adaptive immune system is profoundly involved in the pathogenesis of LN as manifested by B-cell dysregulation and aberrant T-lymphocyte immunity [3,41,42]. In brief, autoreactive B cells produce autoantibodies (such as anti-dsDNA), which form ICs that are deposited in the glomeruli, triggering complement activation and inflammatory cell recruitment. CD4^+^ T cells, particularly the Th1 and Th17 subsets, release proinflammatory cytokines (e.g., IFN-γ, IL-17), further amplifying kidney inflammation. Dysfunction of regulatory T cells is also involved in the impairment of immune tolerance. Taken together, crosstalk between innate and adaptive immunity creates a self-perpetuating loop of inflammation and tissue damage, which underlies the progression of LN. In particular, we recognize that aberrant T-lymphocyte immunity interacts erroneously with B-cell dysregulation in a more sophisticated way. Accordingly, we first highlighted the dissimilarity between PLN and MLN based on T lymphocyte-related abnormalities, then reviewed previous reports regarding aberrant T-lymphocyte immunity among the two different types of LN.

To date, several studies have elucidated differences in T-cell abnormalities between PLN and MLN. Intriguingly, some of those previous reports have indicated that histological variance in LN and two polar morphological forms in the same disease could be due to imbalances in Th cell-related cytokines [16,17,18,19,20]. Japanese clinical studies have demonstrated similar results, with the balance of Th cell-related cytokines tending to show Th1-dominance such as with IFN-γ in PLN, in contrast to Th2-dominance such as with IL-4 in MLN [3,16,21]. In particular, MLN patients without severe hypocomplementemia and elevated auto-antibodies showed apparent Th2-dominance [16,21]. Moreover, in a study analyzing kidney-infiltrating T cells in patients with PLN and MLN using a laser microdissection method, glomerulus-infiltrating T cells in PLN exhibited Th1 and Th17 cytokine profiles, whereas MLN patients showed a Th2 profile [43]. In patients with PLN, activated Th1 and Th17 subsets lead to the production of proinflammatory cytokines like IFN-γ and IL-17, which contribute to glomerular injury by enhancing macrophage activation and neutrophil recruitment [20,44]. Further, Treg dysfunction is frequently reported in PLN and is recognized to lead to a loss of immune tolerance and persistence of autoreactive T cells [45,46]. A study involving 405 patients with biopsy-proven PLN revealed that lower Treg counts in peripheral blood were significantly associated with severe renal outcomes [45]. Impaired production of CD4^+^CD25^+^Foxp3^+^ Tregs in PLN was demonstrated to play a reciprocal role with some cytokines in affecting disease activity and renal damage [46]. In contrast, no clear evidence regarding the association of Tregs with the development of MLN has been established. Previous articles have indicated the plausibility of the T-cell profile in MLN generally being less inflammatory than that in PLN [16,20]. Conversely, as a notable abnormality in T-cell subsets among MLN patients, alterations in T follicular helper (Tfh) cells that support B-cell activation were demonstrated [47]. We consider that the role of the T-cell profile in MLN appears more supportive of autoantibody production than direct mediation of inflammatory damage in renal tissue.

In summary, PLN appears more associated with inflammatory and cytotoxic T-cell responses, including Th1 and Th17 dominance and impaired Treg function, whereas MLN shows a relatively subdued T-cell profile with Th2 dominance and a greater emphasis on humoral mechanisms mediated via B and Tfh cells.

## 5. Dissimilarities Between PLN and MLN Based on Involvement of NETosis

As mentioned, innate and adaptive immune responses are deeply associated with the developmental mechanisms for LN. In this section, we propose the importance of abuse NETosis in the innate system.

In general, formation of NETs is a self-defense reaction, and the process of formation is referred to as NETosis [48,49]. As part of the mechanisms underlying NETosis, generation of reactive oxygen species (ROS) via nicotinamide adenine dinucleotide phosphate (NADPH) oxidase is indispensable [49]. Phagocyte NADPH oxidase (NOX2) consists of five subunits, including gp91^phox^ [50], and exogeneous stimulation via TLRs induces NOX2-dependent ROS generation, which in turn produces bactericidal lytic NETs [49,51]. However, outburst NETosis via excess ROS is considered to induce tissue injury and prolonged inflammation [52,53]. Indeed, in patients with SLE, the involvement of NETs in the pathogenesis has been highlighted [23,24,25]. Persistently activated NETs without proper control are considered to induce LN [25]. More specifically, NETs have been reported as abundant but inadequately degraded in SLE due to the presence of antibodies against NETs themselves and inhibitors of DNAase-1 that prevent NET disassembly [3,54]. NETs themselves, as self-antigens, have been demonstrated to induce the production of type I IFNs by plasmacytoid dendritic cells followed by presentation to autoreactive T and B lymphocytes [3]. In addition, accumulated data have indicated deteriorating effects of NETs on small vessels such as glomerular capillaries [38,54,55]. NETs themselves and autoantibodies against NETs have been demonstrated to induce C1q deposition on NETs, leading to further inhibition of DNase-1 and enhancement of impaired inflammation [56].

Moreover, aside from the above-mentioned lytic NETs, another type of NET has recently garnered attention in the field of basic research. These atypical NETs are released without cell lysis, and are termed “vital NETs” in contrast to the usual lytic NETs that are known as “suicidal NETs” [26]. Intriguingly, these two forms of NETs differ not only in the mechanisms of formation and morphological characteristics, but also in the prevalence across various autoimmune diseases and neutrophil subpopulations. However, the molecular mechanisms underlying each type of NET formation have not been fully elucidated [57,58,59]. Notably, vital NETs have been demonstrated to contain oxidized mitochondrial DNA (mtDNA) and are more frequently observed in patients with LN compared to those with ANCA-associated glomerulonephritis, where suicidal NETs predominate [60]. Vital NETosis occurs faster than suicidal NETosis, and the pathways are assumed to be NOX2-independent [26,61]. Neutrophils primed by several stimuli such as activated platelets, ICs, and calcium ionophores have been demonstrated to form vital NETs with the release of mtDNA [62]. Further, according to previous studies, induction of vital NETosis can significantly enhance inflammatory responses, contributing to the development of autoimmune diseases [59,63]. Importantly, vital NETs in autoimmune diseases are reportedly enriched in oxidized mtDNA instead of nuclear chromatin, which is known to act as a potent inducer of inflammation [63]. Further, unlike suicidal NETs, non-lytic vital NETs are frequently formed in low-density granulocytes (LDGs) during in vitro experiments [23,26]. LDGs have been shown to produce elevated levels of proinflammatory cytokines and are more prone to spontaneously forming NETs compared to normal-density granulocytes [55]. However, most current studies on NET induction have been performed using in vitro or ex vivo models. Further investigations using both experimental animal models and clinical samples from humans with SLE are necessary to better understand the roles and mechanisms of vital NETs in LN.

In terms of associations of the commonly described usual NETs with LN, infiltration of neutrophils into inflamed glomeruli has been regarded as an index of activity for LN in the ISN/RPS classification [15]. The presence of neutrophils in renal biopsy specimens appeared to correlate significantly with active LN [54,55,64,65]. In addition, enrichment of NETs in the peripheral blood of patients with active PLN was indicated by computational analysis [66]. Immunohistochemical and IF analyses using renal biopsy specimens identified the presence of usual NETs showing positivity for citrullinated histone H3 colocalized with myeloperoxidase in the damaged glomeruli of patients with PLN [67]. Moreover, NETs might be involved in the coexistence of PLN and APSN. Emerging evidence suggests that several pathogenic mechanisms underlying APS may also be operative in PLN [56,68,69], highlighting potential shared inflammatory pathways between the two conditions. One such shared mechanism involves the activation of the complement cascade, particularly the generation of complement component C5a, which interacts with its receptor C5aR on neutrophils to promote the release of NETs [56,69]. NETs have been shown to contribute to vascular injury, endothelial dysfunction, and thrombus formation via C5a–C5aR signaling on neutrophils [69,70,71], and these features are hallmarks of APS and may play a critical role in the pathogenesis of APSN in the context of PLN. Collectively, those previous analyses suggested that circulating or infiltrating NET-derived inflammatory peptides and cytokines might contribute to the development of active PLN. We consider that inflamed kidneys may be particularly vulnerable to neutrophil-mediated inflammation.

As described above, the involvement of NETosis in the development of PLN has become apparent from previous reports. However, the influence of NETosis on MLN remains unclear. To the best of our knowledge, direct evidence linking NETosis to the pathogenesis or developmental mechanisms of MLN is currently lacking. Therefore, further investigation with a larger sample size and baseline-matched conditions is warranted to better define phenotypic distinctions in NETosis induction between the PLN group and the MLN group, which may provide insights into the underlying mechanisms for PLN and MLN.

## 6. Dissimilarities Between PLN and MLN Based on Glomerular Expressions of Exostosin 1 and 2

In 2019, the two proteins EXT1/EXT2 on the GBM were highlighted as novel putative antigens in autoimmune disease-related membranous nephropathy [22]. In a report using proteomics and immunohistochemistry, significant expressions of both EXT1/EXT2 were detected together on the GBM of PLA2R-negative secondary membranous nephropathy including MLN, whereas both EXT1/EXT2 were absent in all included cases of PLA2R-associated primary membranous nephropathy [22]. According to recent reports, EXT proteins are stored in the Golgi apparatus of podocytes, and truncated EXT proteins might be secreted from podocytes to the GBM side to affect the development of membranous nephropathy [21,22]. With regard to the prevalence of glomerular EXT1/EXT2-positivity among patients with MLN, 35.0–44.4% were positive according to immunohistochemical investigations of Western populations [22,27,72]. Similarly, expression rates from immunohistochemical investigations were significant among patients with MLN in a Japanese clinical study [73]. Further, 10 of 11 Japanese patients with MLN (90.9%) showed bright granular GBM staining for EXT1/EXT2 on IF analysis [21]. Intriguingly, glomerular EXT1/EXT2 expressions have the potential to offer a predictor of MLN. In one previous cohort [22], two cases initially diagnosed with weak primary membranous nephropathy and partial positivity for EXT1/EXT2 were negative for PLA2R on glomeruli in the first renal biopsy, but by around 6–10 years later, both cases had developed full-blown clinical lupus, and glomerular EXT1/EXT2 staining by IF was strongly positive on the second or third biopsy. Moreover, our colleagues also reported the clinical value of EXT1/EXT2 expressions on glomeruli in IF analysis for identifying the complication of MLN [74] and for diagnosing latent progression of MLN during the postpartum period [75]. Therefore, glomerular EXT1/EXT2 expressions could provide an essential clue for diagnosing both MLN complicated with SLE and early-stage MLN or latently progressive MLN, even before the diagnostic criteria for SLE are fulfilled.

To date, few reports have compared glomerular EXT1/EXT2 expressions between patients with PLN and MLN. Our previous observational study of 11 patients with MLN and 17 patients with PLN investigated whether staining patterns of IF for glomerular EXT1/EXT2 differed between MLN and PLN [21]. Positivity for both EXT1 and EXT2 was higher in the MLN group than in the PLN group. MLN cases demonstrated global and bright granular EXT1/EXT2 expressions on the GBM, whereas PLN cases exhibited segmental, moderately intense expressions on the GBM [21]. On the other hand, Sethi et al. reported that 8 of 18 patients (44.4%) with pure class V LN exhibited bright granular GBM staining for EXT1/EXT2, while 2 of 14 patients (14.3%) with mixed class LN (class III/IV + V) also demonstrated GBM staining for EXT1/EXT2 [22]. In addition, a recent report by the same research group found that 92 of 263 patients (35.0%) with pure class V LN and 30 of 111 patients (27.2%) with mixed-class V LN showed positive EXT1/EXT2 staining along the GBM [27]. Both recent reports emphasized that patients with pure proliferative LN (solely class II, III, or IV) without a membranous component exhibited negative staining for EXT1/EXT2, and EXT1/EXT2 expression was absent in the mesangial area of LN [22,27]. Taken together, these findings indicate that glomerular EXT1/EXT2 staining patterns might differ between patients with MLN and PLN, and glomerular EXT1/EXT2 could be a causative factor in the development of MLN.

Our analysis demonstrates intriguing findings. Glomerular EXT1/EXT2 positivity was not associated with the degree of proteinuria or renal dysfunction in patients with MLN. However, patients with EXT1/EXT2-positive MLN exhibited lower levels of serum anti-dsDNA Ab and circulating ICs compared to patients with EXT1/EXT2-negative PLN [21]. Further, serum complement levels and IL-4/IFN-γ ratios were higher in EXT1/EXT2-positive MLN than in EXT1/EXT2-negative PLN [21]. In short, IF staining for glomerular EXT1/EXT2 might be associated with characteristic features in MLN. Glomerular EXT1/EXT2 expression tended to be higher in MLN patients with a Th2-dominant immune profile, in the absence of severe hypocomplementemia and markedly elevated autoantibody levels. According to previous reports [14,16,17], an immunological profile characterized by normal complement levels, normal autoantibody titers, and a Th2-dominant immune response is recognized as a distinctive feature of MLN. Such immunological trends are considered to be linked to the unique pathogenic mechanisms underlying MLN. In general, the pathogenesis of active PLN involves the deposition of circulating ICs in the glomerular capillaries or the binding of autoantibodies to autoantigens (e.g., nucleosomes or C1q) that have become planted within the glomeruli [8,76]. Meanwhile, the pathogenesis of MLN is thought to involve the formation of ICs in situ, where autoantibodies bind directly to endogenous glomerular antigens [14,76,77]. In this context, glomerular EXT1/EXT2 might be involved in the unique developmental mechanisms of MLN. We speculate that EXT1/EXT2 on the GBM may be shed from podocytes in response to in situ ICs containing autoantigens other than EXT1/EXT2. Interestingly, Ravindran et al. also proposed that increased synthesis of glomerular heparan sulfatase, potentially triggered by oversecretion of truncated EXT1/EXT2, may coat ICs on the GBM and attenuate the inflammatory responses induced [27]. Further studies are warranted to confirm this hypothesis and to elucidate the precise roles of glomerular EXT1/EXT2 in the pathogenesis of MLN.

Regarding the critical question of whether treatment strategies and responses differ between PLN and MLN based on the presence or absence of glomerular EXT1/EXT2 expressions, current evidence remains extremely limited. To date, the only relevant cohort analysis is that conducted by Ravindran et al., which assessed renal prognosis in relation to glomerular EXT1/EXT2 expressions [27]. Their study found that patients with LN (class 5 MLN with or without class 3/4 lesions) who were EXT1/EXT2-negative exhibited significantly higher rates of global glomerulosclerosis and interstitial fibrosis/tubular atrophy in kidney biopsy specimens. Moreover, multivariable Cox regression analysis revealed that the absence of glomerular EXT1/EXT2 expressions significantly increased the hazard ratio for progression to end-stage kidney disease (ESKD) during a 10-year period in this population. In other words, EXT1/EXT2-negative patients progressed to ESKD more rapidly and more frequently than those with positive expressions [27]. These findings suggest that glomerular EXT1/EXT2 expressions may serve as an indicator for treatment response and long-term renal prognosis in patients with PLN and MLN. However, to validate these observations, further large-scale, multiethnic observational studies are needed.

## 7. Concluding Remarks

This review has described the characteristic features of PLN and MLN, with a focus on highlighting the distinct pathogenesis of these two major subtypes of LN. Figure 1 provides a simple summary of the dissimilarities between the two types of LN according to typical immunoserological features, T lymphocyte-related abnormalities, involvement of NETs, and glomerular EXT1/EXT2 expressions. In addition, Figure 2 provides a graphical summary of our proposed concept. In PLN, excessive NETosis and pro-inflammatory T-cell responses, particularly those involving Th1 and Th17 cells, promote the deposition of circulating ICs within glomeruli. In contrast, MLN may develop under a Th2-skewed immune environment, which might be related to in situ formation of ICs in glomeruli. Glomerular ICs are not derived from systemic circulation but instead form locally on the GBM, consisting of EXT1/EXT2 proteins shed from podocytes and their corresponding autoantibodies. Unfortunately, current treatment strategies are not based on differences in development mechanisms between PLN and MLN. Instead, therapeutic decisions are often based solely on the histological classification of LN. Further research to elucidate the pathogenic pathways underlying these two polar morphological forms of the same disease is thus warranted.

## Figures and Tables

**Figure 1 ijms-26-08769-f001:**
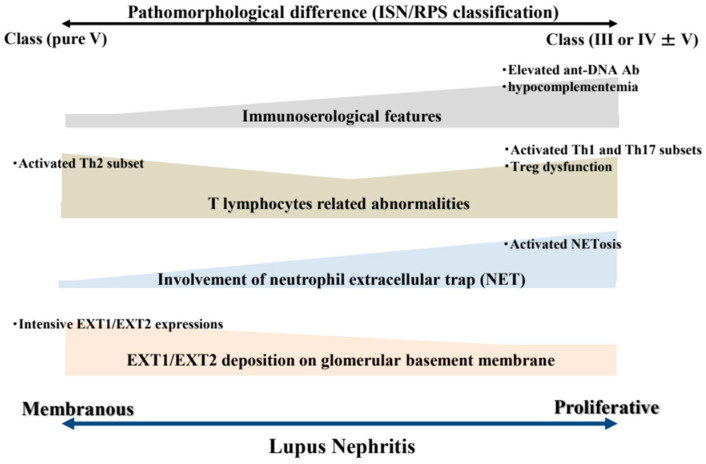
Summary of the distinction between PLN and MLN. The dissimilarity between the two different types of lupus nephritis might be attributable to pathological findings, immunoserological features based on the presence or absence of serum low complement or anti-DNA antibodies, T lymphocyte-related abnormalities, involvement of neutrophil extracellular traps, and glomerular EXT1/EXT2 expressions. Abbreviations: Ab, antibody; ISN/RPS, International Society of Nephrology/Renal Pathology Society; Th, T-helper; EXT1/EXT2, exostosin 1 and exostosin 2.

**Figure 2 ijms-26-08769-f002:**
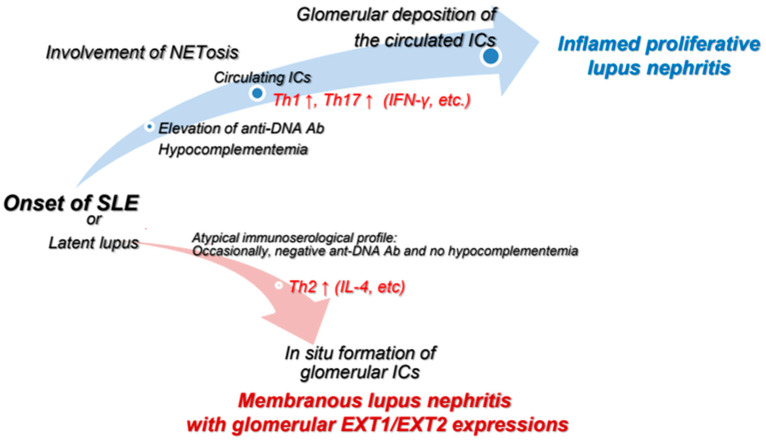
Schema of hypothetical mechanisms underlying the development of inflamed PLN and EXT1/EXT2-related MLN. In PLN, excess NETosis and pro-inflammatory T-cell responses, particularly involving activated Th1 and Th17 cells, contribute to the deposition of circulating ICs within glomeruli. In contrast, EXT1/EXT2-related MLN may arise under a Th2-dominant immune environment, which promotes in situ formation of ICs. These ICs might not be derived from the circulation, but rather form locally on the GBM. Abbreviations: Ab, antibody; Th, T-helper; ICs, immunocomplexes; EXT1/EXT2, exostosin 1 and exostosin 2.

## Data Availability

No new data were created or analyzed in this study. Data sharing is not applicable to this article.

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
