# Peer review of "Distinction Between Proliferative Lupus Nephritis and Membranous Lupus Nephritis Based on Inflammation, NETosis, and Glomerular Exostosin"

_ijms, 2025, doi:10.3390/ijms26188769_

Round 1

Reviewer 1 Report

Comments and Suggestions for Authors

The authors attempt to summarize and contextualize current knowledge regarding the pathological and immunological distinctions between proliferative and membranous lupus nephritis, with a focus on NETosis and EXT1/EXT2 expression. While this aim is timely and relevant, some important points have to be taken into account.

My main concern is that the Methodology and Search Strategy for the review are not clearly and sufficiently presented. In the "Method for Literature Search" section, the strategy used for combining search terms is unclear. Please specify whether Boolean operators were applied (e.g., AND, OR) and in what combinations. The overall methodology for the literature review lacks sufficient detail. The authors should clarify how many records were retrieved from the initial search, how many were excluded (and why), and how many were finally included in the review. Consider presenting this in a flow diagram format for transparency and reproducibility.

In addition, the rationale for focusing on exostosin (EXT1/EXT2) as a key marker for distinguishing lupus nephritis profiles should be further justified, as its added value is not clear for the management of LN. Why is exostosin considered particularly promising compared to other molecular profiles?

The review currently does not provide adequate discussion on how the described differences in histological or molecular features account for differential treatment responses in LN. A more detailed analysis in this regard would add depth and clinical relevance to the paper.

Minor Point

Please improve the quality and resolution of the figures, especially Figure 1, as the current version is difficult to read.

Author Response

Reviewer 1

The authors attempt to summarize and contextualize current knowledge regarding the pathological and immunological distinctions between proliferative and membranous lupus nephritis, with a focus on NETosis and EXT1/EXT2 expression. While this aim is timely and relevant, some important points have to be taken into account.

My main concern is that the Methodology and Search Strategy for the review are not clearly and sufficiently presented. In the "Method for Literature Search" section, the strategy used for combining search terms is unclear. Please specify whether Boolean operators were applied (e.g., AND, OR) and in what combinations. The overall methodology for the literature review lacks sufficient detail. The authors should clarify how many records were retrieved from the initial search, how many were excluded (and why), and how many were finally included in the review. Consider presenting this in a flow diagram format for transparency and reproducibility.

Thank you for your suggestion which is very important for this review article. Actually, we first performed a comprehensive search using the term "Lupus nephritis," which yielded 9,040 articles. Subsequently, we refined our search by combining "Lupus nephritis" with specific keywords, resulting in 248 articles for "Lupus nephritis, inflammation, and lymphocyte," 114 articles for "Lupus nephritis and membranous nephropathy," 67 articles for "Lupus nephritis and neutrophil extracellular traps," and 14 articles for "Lupus nephritis and exostosin." From these, we selected papers of particular relevance and significance to be cited and discussed in this review article. We modified the manuscript in alignment with this search strategy (Page 2, Line 92-Page 3, Line 103).

In addition, the rationale for focusing on exostosin (EXT1/EXT2) as a key marker for distinguishing lupus nephritis profiles should be further justified, as its added value is not clear for the management of LN. Why is exostosin considered particularly promising compared to other molecular profiles?

    Thank you for your valuable suggestion. As described in the abstract (Page 1, Lines 24–25), the introduction (Page 2, Lines 67–68), and the section on pathomorphological differences between PLN and MLN (Page 4, Lines 179–181), patients with MLN often do not exhibit the typical immunoserological features of SLE, such as low serum complement levels or elevated anti-DNA antibody titers.

In clinical practice, diagnosing MLN in the context of SLE can be challenging when such atypical immunoserological features are present. In particular, it is often difficult to distinguish MLN from primary membranous nephropathy (PMN). Therefore, identifying disease-specific or pathogenic antigens expressed on the glomerular epithelium may provide important clues to the pathogenesis of MLN. For example, M-type phospholipase A2 receptor (PLA2R), a known antigen for PMN, was mentioned in the manuscript (Page 4, Lines 181–183).

Furthermore, in the section discussing dissimilarities between PLN and MLN based on EXT1/EXT2 expression, we noted that glomerular EXT1/EXT2 expression may serve as a potential indicator for predicting MLN (Page 7, Lines 338–344). Taken together, we believe that our current findings regarding glomerular EXT1/EXT2 expressions in patients with lupus nephritis may provide valuable insights for renal diseases physicians. In particular, such information may aid in the diagnosis of not only MLN in patients with established SLE but also cases of incipient MLN that have not yet fulfilled the diagnostic criteria for SLE. Accordingly, we have added further explanation on this point in the revised manuscript (Page 7, Lines 346–349) in response to the reviewer’s suggestion.

The review currently does not provide adequate discussion on how the described differences in histological or molecular features account for differential treatment responses in LN. A more detailed analysis in this regard would add depth and clinical relevance to the paper.

In this article, we focused on exploring the developmental mechanisms underlying PLN and MLN, rather than comparing treatment strategies between the two groups. As you correctly pointed out, discussing differences in treatment strategies and responses is indeed important and valuable for both clinicians and future research. However, current evidence regarding treatment methods or responses specific to PLN and MLN remains extremely limited. Since the pathogenic mechanisms of these two conditions have not been clearly elucidated in the first place, we deliberately refrained from discussing treatment approaches, as such discussion would be premature and potentially misleading.

However, to our knowledge, the only relevant study is a cohort analysis by Ravindran et al., which evaluated renal prognosis based on the presence or absence of glomerular EXT1/EXT2 expression. They reported that patients with lupus nephritis who were EXT1/EXT2-negative had significantly higher levels of global glomerulosclerosis and interstitial fibrosis/tubular atrophy (IFTA) in kidney biopsy specimens. Moreover, multivariable Cox regression analysis revealed that absence of EXT1/EXT2 expression significantly increased the hazard ratio for progression to end-stage kidney disease (ESKD) at 10 years. In other word, EXT1/EXT2-negative patients progressed to ESKD more rapidly and more frequently than those with positive expression. While part of these findings was mentioned in the initial version of our manuscript, we have now added further details regarding this study. Additionally, we have included a comment on the lack of evidence concerning treatment strategies and responses as a limitation of this article (Page 9, Line 394-408).

Minor Point

Please improve the quality and resolution of the figures, especially Figure 1, as the current version is difficult to read.

We address to the issue. You suggest to improve the quality and resolution of the figures, especially Figure 1, as the current version is difficult to interpret clearly. We made effort to improve the quality of Figure 1 and 2 as much as we can.

Regarding Figure 1, we acknowledge that the illustration may appear somewhat abstract. However, our intention was to summarize and contrast the clinical, pathological, immunological, NETosis-related, and EXT-related differences between PLN and MLN. As mentioned in the main text, these factors tend to show characteristic patterns in each group; however, they cannot be categorized in a simple binary manner (i.e., presence or absence, all-or-nothing). Rather, we perceive a shift in a delicate balance, much like a seesaw, where the equilibrium between these elements is disrupted differently in PLN and MLN. Therefore, we hope you will understand our rationale for presenting the figure in this format.

Reviewer 2 Report

Comments and Suggestions for Authors

Thank you for the opportunity to review this manuscript.

Overall, the paper is well written and offers a balanced and comprehensive literature review of the pathophysiological mechanisms distinguishing proliferative lupus nephritis (PLN) from membranous lupus nephritis (MLN). The discussion is well-structured and provides a valuable synthesis of current evidence.

I only have minor comments:

In addition to systemic lupus erythematosus (SLE), neutrophil extracellular traps (NETs) have been implicated in other autoimmune renal diseases, most notably antiphospholipid syndrome (APS), which coexists with SLE in approximately 30–40% of cases. Although the current literature on this association is limited, I suggest that the authors discuss the potential impact of antiphospholipid antibody (aPL) positivity and the possible coexistence of PLN with aPL nephropathy, as defined by the ACR/EULAR pathology.

Moreover, several pathogenic mechanisms described in APS may also be relevant to lupus nephritis. For example, C5a-mediated stimulation of NET formation via C5aR may represent a shared pathway worthy of discussion.

Author Response

Reviewer 2

Thank you for the opportunity to review this manuscript.

Overall, the paper is well written and offers a balanced and comprehensive literature review of the pathophysiological mechanisms distinguishing proliferative lupus nephritis (PLN) from membranous lupus nephritis (MLN). The discussion is well-structured and provides a valuable synthesis of current evidence.

I only have minor comments:

In addition to systemic lupus erythematosus (SLE), neutrophil extracellular traps (NETs) have been implicated in other autoimmune renal diseases, most notably antiphospholipid syndrome (APS), which coexists with SLE in approximately 30–40% of cases. Although the current literature on this association is limited, I suggest that the authors discuss the potential impact of antiphospholipid antibody (aPL) positivity and the possible coexistence of PLN with aPL nephropathy, as defined by the ACR/EULAR pathology.

Moreover, several pathogenic mechanisms described in APS may also be relevant to lupus nephritis. For example, C5a-mediated stimulation of NET formation via C5aR may represent a shared pathway worthy of discussion.

Thank you very much for your valuable suggestion, which has significantly enhanced the quality of our review article. In response, we have included additional information regarding the association between antiphospholipid syndrome nephritis and proliferative lupus nephritis, as per your recommendation (Page 4, Line 151-167) (Page 6, Line 301-310).

Round 2

Reviewer 1 Report

Comments and Suggestions for Authors

The reply and the additions of the authors are helpful.

However, the paragraph for the literature review is still vague and does not allow the reproducibility of the search. It reports broad keyword combinations and a 2010–2025 window, but it does not identify the databases/platforms searched, the final search date or the exact search strategies (Boolean strings, field tags, MeSH/Emtree terms). The selection approach remains subjective (“reports of particular relevance and significance”) with no predefined inclusion/exclusion criteria, no description of the screening workflow (number of reviewers, deduplication, how disagreements were resolved), and no accounting of records at each stage. 

Please provide the missing methodological details and ideally the full search strategies.

Author Response

August 30, 2025

Dear Editor-in-Chief, International Journal of Molecular Science

Manuscript ID: ijms-3806658

Distinction between proliferative lupus nephritis and membranous lupus nephritis based on inflammation, NETosis, and glomerular exostosin

We appreciate the meticulous review of our manuscript. We welcome the opportunity to submit a revised manuscript. All changes are marked up using the “red color” in the word file which is named revised manuscript with track change. In addition, complete final manuscript after revision was also submitted as a word file named revised manuscript.

We have addressed the concerns raised as follows:

Reviewer 1

The reply and the additions of the authors are helpful.

However, the paragraph for the literature review is still vague and does not allow the reproducibility of the search. It reports broad keyword combinations and a 2010–2025 window, but it does not identify the databases/platforms searched, the final search date or the exact search strategies (Boolean strings, field tags, MeSH/Emtree terms). The selection approach remains subjective (“reports of particular relevance and significance”) with no predefined inclusion/exclusion criteria, no description of the screening workflow (number of reviewers, deduplication, how disagreements were resolved), and no accounting of records at each stage.

Please provide the missing methodological details and ideally the full search strategies.

Thank you again for your constructive and detailed feedback. We fully acknowledge the importance of providing a transparent and reproducible methodology for literature reviews. In response, we have substantially revised the “Method for Literature Search” section to address the concerns you raised. The revised version now includes the following details:

  • Databases Searched: We conducted our literature search using PubMed (MEDLINE).

  • Search Period: The search covered literature published between January 1, 2010, and July 31, 2025.

  • Search Strategy: The full search strategy for PubMed was as follows:

("Lupus" [MeSH Terms] OR "lupus" [Title/Abstract]) AND ("inflammation" [Title/Abstract] OR "lymphocyte Title/Abstract])

Additional searches were conducted using combinations such as:

"lupus" AND "membranous nephropathy"

"lupus" AND "neutrophil extracellular traps"

"lupus" AND "exostosin".

Although we previously described 'lupus nephritis' as the key search term in the manuscript and response letter, we would like to correct this, as the actual search was conducted using the broader term 'lupus'

  • Deduplication: Duplicate records were removed using the EndNote reference management software.

  • Screening: The initial literature search and screening of titles and abstracts were primarily conducted by the first author. However, the selection of relevant studies and evaluation of their content were rigorously assessed in collaboration with the co-authors to ensure accuracy and objectivity. Discrepancies or uncertainties regarding study inclusion were discussed and resolved through consensus among the authors. Furthermore, during the selection and evaluation process, studies published prior to January 1, 2010 were also considered if they were deemed highly reliable and valuable this review. Such references were incorporated with the agreement of all co-authors and included in the final manuscript.

  • Eligibility Criteria:

Inclusion: Original peer-reviewed articles in English, human studies, and articles focusing on the pathophysiology, biomarkers, or treatment strategies related to lupus nephritis. Additionally, case reports and review articles were considered for inclusion if they were judged particularly valuable to the context of the review. These were included with the agreement of all co-authors.

Exclusion: Conference abstracts and non-English publications were excluded.

  • PRISMA Flow Diagram: We fully acknowledge that, ideally, a PRISMA-compliant flow diagram should be included to enhance transparency and reproducibility. However, this review is not a systematic review of clinical trials or treatment guidelines, nor does it primarily focus on the evaluation of clinical data. Given the narrative nature of this review—which integrates a broad range of literature, including basic and translational research—we determined that a detailed PRISMA diagram may not be essential in this context. We also recognize that constructing such a diagram in a methodologically rigorous manner posed practical challenges for our team. Nevertheless, we have conducted the literature search and selection process with diligence and integrity, as outlined in the revised methodology. We sincerely hope that our efforts to present a transparent and honest account of the review process will be recognized in lieu of a formal flow diagram.

These additions aim to enhance the clarity, transparency, and reproducibility of our review methodology. We are grateful for your insightful comments, which have significantly strengthened the rigor and quality of our manuscript. We hope that the revised version will now be considered suitable for publication in the International Journal of Molecular Sciences.

We look forward to receiving your response.

Sincerely,

Yukihiro Wada

Department of Nephrology, Kitasato University School of Medicine

E-mail address: wada-y@kitasato-u.ac.jp

Round 3

Reviewer 1 Report

Comments and Suggestions for Authors

Thank you for your reply. I have no further comments.